# Endogenic Phenolic Compounds of Barley as Potential Biomarkers Related to Grain Mycotoxin Production and Cultivar Selection

**DOI:** 10.3390/biology12101306

**Published:** 2023-10-03

**Authors:** Ana Badea, James R. Tucker, Ali Sabra, Thomas Netticadan, Barbara Blackwell, Liping Yu, Chamali Kodikara, Champa Wijekoon

**Affiliations:** 1Agriculture and Agri-Food Canada, Brandon Research and Development Centre, Brandon, MB R7A 5Y3, Canada; ana.badea@agr.gc.ca (A.B.); james.tucker@agr.gc.ca (J.R.T.); 2Agriculture and Agri-Food Canada, Morden Research and Development Centre, Morden, MB R6M 1Y5, Canada; ali.sabra@agr.gc.ca (A.S.); thomas.netticadan@agr.gc.ca (T.N.); liping.yu@agr.gc.ca (L.Y.); chamali.kodikara@agr.gc.ca (C.K.); 3Canadian Centre for Agri-Food Research in Health and Medicine, Winnipeg, MB R3C 1B2, Canada; 4Agriculture and Agri-Food Canada, Ottawa Research and Development Centre, Ottawa, ON K1A 0C6, Canada; barbara.blackwell@agr.gc.ca

**Keywords:** fusarium head blight (FHB), phenolic compounds, deoxynivalenol (DON), barley, cultivars

## Abstract

**Simple Summary:**

Barley is the fourth largest cereal crop in the world. Fusarium head blight (FHB) is one of the diseases in barley producing mycotoxins, such as deoxynivalenol (DON), that could affect grain quality as well as human and animal health worldwide. Due to limited reliable biomarkers to identify and develop FHB-resistant cultivars in barley, we investigated the composition of phenolic compounds in ten barley cultivars under clean and FHB-infected conditions. We analyzed free and bound forms of phenolic compounds and identified differences among tested cultivars. Analysis of mycotoxin DON content showed that resistant cultivars produced less compared to susceptible cultivars. In addition, the resistant cultivars showed higher amounts of major phenolic compounds compared to the known susceptible cultivar. The results of this study suggest that phenolic compounds in barley could have a role as potential biomarkers to identify and develop FHB-resistant barley cultivars.

**Abstract:**

Barley (*Hordeum vulgare* L.) is the fourth largest cereal crop in the world. One of the most devastating diseases in barley worldwide is Fusarium head blight (FHB) caused by *Fusarium graminearum* Schwabe. Several mycotoxins are produced by FHB infection, and deoxynivalenol (DON) is one of them responsible for the deterioration of grain quality. The current limited number of reliable molecular markers makes the development of FHB-resistant cultivars rather difficult and laborious. Moreover, there is a limited number of designed specific biomarkers that could distinguish the FHB resistance and mycotoxin accumulation in barley cultivars. This study investigated the phenolic compounds of ten different Canadian barley cultivars, grown in artificially FHB-infected and non-infected field trials. The enzyme-linked immunosorbent assay (ELISA) was used to assess the presence of DON in the harvested infected grains of each tested variety. High-performance liquid chromatography (HPLC) analysis was performed using both infected and non-infected samples. We identified differences among cultivars tested in non-infected samples through quantitative analysis of free and bound phenolic compounds. The resistant cultivars showed higher amounts of major bound phenolic compounds compared to the susceptible check CDC Bold. Additionally, the FHB-infected cultivars produced significantly higher amounts of sinapic acid (SIN) () and catechin (CAT) in the soluble free form of phenolics in barley compared to the non-infected subjects. This study suggests that phenolic compounds in barley could allow barley breeders to precisely identify and develop FHB-resistant barley germplasm and cultivars.

## 1. Introduction

Plants are constantly facing a wide range of biotic (e.g.. fungi, bacteria and pests) and abiotic (e.g., drought and waterlogging) stresses. Fusarium head blight (FHB) in cereal plants is one of the most serious biotic stresses worldwide including in western Canada. This disease not only causes a loss in grain yield, but also a deterioration in grain quality, by producing several mycotoxins such as deoxynivalenol (DON) that could affect human and animal health [1]. There is a need to understand and capitalize on the various plant defense mechanisms in order to adapt to the potential increase in the exiting stresses and/or emergence of stresses under the changing climate.

Barley (*Hordeum vulgare* L.), an ancient grain which has been used for thousands of years, is currently the fourth largest cereal crop worldwide. Production of barley and its value-added products impact various industries including breweries, food processors, feed mills and livestock operations. Barley grain is rich in antioxidant phytochemicals, fibre, fatty acids, proteins, phenolic compounds, vitamins and minerals [2]. 

Primary and secondary metabolites are produced by the plants. While the primary metabolites promote growth, reproduction and development, the secondary metabolites play an important role in the survival of a plant in its environment [3]. A large portion of a plant’s secondary metabolites are represented by phenolic compounds, which are involved in the defense mechanism of the plant. For example, previous studies showed the involvement of phenolic compounds in the FHB resistance of barley. Phenylpropanoid pathway metabolites such as phenylalanine and para-coumaric acid (PCA) showed a two-fold or greater abundance in FHB resistant vs. susceptible lines [4]. The majority of phenolic compounds are bound to the cell walls of cereals [5]. Two forms of phenolic compounds (free and bound) are found in cereals, where the free form includes either free acids or esterified to sugar conjugates and the bound form involves conjugated insoluble phenolic compounds to several polysaccharides and to lignin through ester and ether bonds [6]. The soluble forms of cereal phenolic compounds are believed to be compartmentalized in the vacuoles, whereas the insoluble forms are incorporated in cell walls [7]. Unlike pesticides, phenolic compounds are naturally occurring phytochemicals that are widely recognized for numerous human health benefits [8].

Several phenolic compounds are involved in modulating the production of mycotoxins in vitro in *Fusarium* species despite their effect being highly variable based on the class of fungal species, mycotoxins and the experimental conditions [9]. Phenylpropanoids such as syringic acid and sinapic acid (SIN) derivatives were previously reported as FHB-resistance-related metabolites in double haploid barley lines [10]. More recently, a study reported a potential role in flavonoid and hydroxycinnamic acid amides on the resistance against FHB in an FHB-resistant cultivar/genotype [11]. Out of the phenolic compounds, ferulic acid (FA) is one of the most abundant in cereals and reported to inhibit fungal growth in *Fusarium graminearum* [12]. In addition, the concentrations of FA were shown to be negatively correlated with the relative ratings of cultivars concerning FHB and DON contents in wheat [13]. It is believed that cinnamic acid derivatives, such as para-coumaric acid (PCA), caffeic acid (CAF), isoferulic acid (Iso FA), SIN and FA accumulated in the kernel may be the contributors to FHB resistance [6]. However, the role of phenolic compounds on FHB resistance in Canadian barley genotypes/cultivars is still unknown. Recent metabolomics tools have greatly facilitated food and nutrition research [14]. In this study, we investigated the phenolic and mycotoxin composition of barley cultivars to uncover potential biomarkers related to FHB that could help in the development of ‘climate-smart’ barley that will have in-built resistance to FHB infection.

## 2. Materials and Methods

### 2.1. Barley Grain Samples

Powdered grains of 10 two-row spring hulled barley cultivars, of malting and general purpose, including AAC Goldman, AC Metcalfe, AAC Synergy, CDC Bold, CDC Bow, CDC Copeland, CDC Mindon, Harrington, Lowe and Newdale were used for this study (Appendix A). The cultivars were selected specifically for a range of reactions to FHB. Clean and FHB-infected grains were used. Clean grains were generated from a field trial grown during the 2020 crop season at the Agriculture and Agri-Food Canada, Brandon Research and Development Centre (AAFC-BRDC), Brandon, MB, as described by Wijekoon et al. [15].

The FHB-infected grains were generated from the AAFC-BRDC disease field trial during the 2020 crop season. The cultivars were grown in short rows (0.9 m) under a randomized complete block design (*n* = 3). Corn kernels (4 kg) were soaked overnight in a stainless steel pan. Kernels were drained, covered by tin foil and autoclaved at 121 °C for 60 min. A potato dextrose agar media plate fully colonized by *F. graminearum* was added to the corn within a biosafety cabinet, and incubated at room temperature for three weeks. Afterwards, the infected corn was dried by spreading it on corrugated steel trays and subjected to forced air. Grain spawn inoculum, consisting of corn kernels infected with four isolates (50:50 3ADON:15ADON), was spread on the soil surface (5 g/m^2^) between rows, before the flag leaf stage. Inoculum was reapplied at weekly intervals for a total of three applications. Irrigation was applied daily with fine water droplet style sprinkler nozzles (NaanDanJain Irrigation Ltd, CA, USA) between 04:00 and 06:00 h, and then again between 18:00 and 20:00 h. At maturity, grains were harvested by a stationary combine. Grains were milled by a Perten Labmill 3100, PerkinElmer Inc. (Woodbridge, ON, Canada).

### 2.2. Extraction and Analysis of Phenolic Compounds and Antioxidant Assay

Phenolic compound standards (PCA, CAF, SIN, FA, Iso FA, CAT, 4HBA, VAN A) used in this study were purchased from Millipore Sigma (Sigma-Aldrich Canada, Oakville, ON, Canada). Free and bound barley phenolic compounds were extracted and analyzed according to Wijekoon et al. [15]. In brief, ground grains were extracted with ethanol 90% solution at a ratio of 1:9 (solid:liquid) with sonication at 70 °C for 30 min. The supernatant containing the free phenolics was filtered and dried under a vacuum and reconstituted in methanol. The extract was then filtered using a syringe filter of 0.2 µm and stored at −20 °C. Extraction of bound phenolic compounds from barley was based on successive acid and alkaline hydrolysis as previously described by Hajji et al. [16], with modifications. Alkaline hydrolysis was performed under sonication, and the bound phenolics were extracted by ethyl acetate. The extract was reconstituted in methanol and syringe-filtered into an insert vial before HPLC analysis. Analysis was done using HPLC Dionex 3000 Ultimate (Thermo Scientific, Waltham, MA, USA) equipped with a C18 reversed-phase column (Acclaim 120, 4.6 × 250 mm, 5 µm). Compounds were separated through a gradient elution in which the mobile phase A comprised 0.1% phosphoric acid in the water, while the mobile phase B was acetonitrile. The proportion of B was gradually increased with time to allow the separation of compounds of interest in 38 min. The flow rate was 1 ml/min and the injection volume was 20 µl. Different wavelengths were used to detect the phenolic compounds in extracts. For example, cinnamic acid derivatives were quantified at 325 nm, while benzoic acid derivatives and flavonoids were quantified at 280 nm (Figure 1). Quantification was based on an external standard calibration method. Calibration curves were established by preparing serial dilutions of the standards mix with methanol covering the range from 1 to 100 µg/ml, which gave a good linearity with R^2^≥ 0.998. ABTS antioxidant activity analysis following Wijekoon et al. [15] was done on both free and bound phenolic extracts.

### 2.3. Disease Severity and Mycotoxin Analysis

Fusarium head blight was rated on a scale of 0–5 [17], which represents a composite score of both incidence and severity, where 0 = no disease and 5 = high disease with 50% or more spikes infected; up to 50%+ of spike diseased (for full scale, see [17]). Data were analyzed in SAS JMP v 16.2.0, where replicate was considered a random factor. Samples were cleaned to remove chaff and debris using an SLN3 sample cleaner (Pfeuffer GmbH, Kitzingen, Germany). A 20 g sub-sample was removed and ground using an LM 3610 laboratory mill (PerkinElmer Inc., Shelton, CT, USA). A 1.0 g sample was extracted in 10 mL aqueous solution of methanol (10% vol/vol). Deoxynivalenol content was determined via enzyme-linked immunosorbent assay (ELISA) as per Sinha and Savard [18]. Samples that differed by >10% were reanalyzed. 

### 2.4. Statistical Analysis

One-way ANOVA was used to analyze the quantitative data of all cultivars studied by OriginPro 2022 statistical software (OriginLab Corporation, Northampton, MA, USA). Mean values of phenolics were compared among cultivars using Tukey’s test, and the significance was determined at *p* ≤ 0.05. In addition, principal component analysis was performed using Minitab (Minitab LLC) to find the trends in different barley cultivars.

## 3. Results

### 3.1. Assessment of Phenolic Compounds in Clean and FHB-Infected Barley Grains

Free and bound phenolic compounds were quantified in both clean and FHB-infected barley samples. The differences in the chromatograms of the susceptible check (CDC Bold) and the moderately resistant check cultivar (CDC Mindon) are shown in Figure 2. 

PCA, SIN, FA, Iso FA and CAT were detected in the free phenolic fraction. Interestingly, Iso FA was only detected in the free phenolic fraction of the infected plants, while SIN and CAT showed a significant increase in FHB-infected barley cultivars compared with clean cultivars tested (Appendix A). The FHB-susceptible cultivar CDC Bold had the highest CAT content, while the moderately resistant Lowe cultivar had the lowest CAT content among the clean and FHB-infected cultivars tested. Although clean grains of some of the cultivars showed a quantifiable amount of PCA, the FHB-infected barley cultivars did not show any quantifiable PCA contents in the free phenolic fraction (Figure 3A, Appendix A). 

Phenolic compounds in bound barley included PCA, SIN, FA, Iso FA, CAT, CAF, 4HBA and VAN A. Interestingly, clean grains of the susceptible cultivar CDC Bold showed the lowest contents of PCA, CAF, SIN, FA, Iso FA, CAT and VAN A, while the moderately resistant cultivars, such as CDC Mindon and Harrington, showed relatively higher contents of CAF, FA and Iso FA. In the FHB-infected grains, the moderately FHB-resistant Lowe cultivar had the highest contents of most of the phenolic compounds tested when infected (Figure 3B, and Appendix A). Although the bound phenolic contents of many cultivars did not show significant changes after infection with FHB, CDC Bold and CDC Copeland cultivars showed variation in their phenolic content after infection. A pairwise comparison of highly abundant bound PCA and FA contents between clean and infected barley showed that FHB-susceptible CDC Bold had a significant increase in both compounds after infection while the intermediate CDC Copeland cultivar had a significant decrease after infection (Table 1 and Table 2). The antioxidant activity assay was performed to see any changes in clean and FHB-infected barley samples. The antioxidant activity changes were highly variable and not significantly different in free and bound phenolic extracts with or without infection (Appendix A). The sum of the total individual phenolic compounds of free and bound extracts showed that the FHB infection has increased the phenolic compound accumulation in barley grains. For instance, total CAT and SIN contents showed a significant increase in FHB infection in most of the susceptible and moderately resistant barley cultivars (Appendix A).

Phenolic compounds in bound barley include PCA, SA, FA, Iso FA, CAT, caffeic acid (CAF), 4 hydroxybenzoic acids (4HBA) and vanillic acid (VAN A). Interestingly, clean grains of the susceptible cultivar CDC Bold showed the lowest contents of PCA, CAF, SIN, FA, Iso FA, CAT and VAN A, while the moderately resistant cultivars, such as CDC Mindon and Harrington showed relatively higher contents of CAF, FA and Iso FA. In the FHB-infected grains, the moderately FHB-resistant Lowe cultivar had the highest contents of most of the phenolic compounds tested when infected (Figure 3B, Table 2 and Table 3, and Appendix A). Although the bound phenolic contents of many cultivars did not show significant changes after infection with FHB, CDC Bold and CDC Copeland cultivars showed variation in their phenolic content after infection. A pairwise comparison of highly abundant bound PCA and FA contents between clean and infected barley showed that FHB-susceptible CDC Bold had a significant increase in both compounds after infection while the intermediate CDC Copeland cultivar had a significant decrease after infection (Appendix A). The antioxidant activity assay was performed to see any changes in clean and FHB-infected barley samples. The antioxidant activity changes were highly variable and not significantly different in free and bound phenolic extracts with or without infection (Appendix A). The sum of the total individual phenolic compounds of free and bound extracts showed that the FHB infection increased the accumulation of phenolic compounds in barley grains. For instance, total CAT and SIN contents showed a significant increase in FHB infection in most of the susceptible and moderately resistant barley cultivars (Appendix A).

### 3.2. Principal Component Analysis Using Total (Free and Bound) Phenolic Contents of FHB-Infected Barley

The principal component analysis reduced the dimensionality of the data set, showing the summary of the data and more than 74% of the total variations of tested phenolic compounds in barley cultivars. The data variance distributed in the first three variables is because of the high correlation between those phenolic compounds and barley cultivars (Figure 4, Appendix A).

The first principal component explained 36% of the total variance that was contributed mainly by PCA, SIN, 4HBA, VAN A, FA and Iso FA with high positive loadings. The second principal component accounted for 25% of variations due to the large positive loadings from FA and Iso FA, and a small positive loading from SIN. The third principal component accounted for 14% of the total variations, mainly due to CAF, Iso FA and VAN A. The loading plot indicates the importance of PCA, 4HBA and VAN A for the positive loadings of principal component one. FA and Iso FA showed nearly the same direction, which confirms their positive relationship as described in the eigenvectors (Appendix A). The scree plot (Figure 4A) shows two distinct groups compared to the other cultivar distributions. One group consists of two cultivars, AAC Synergy and CDC Mindon, which have similar loadings in the scree plots, likely due to the presence of FA and Iso FA. The other distinct group consists of three cultivars: AC Metcalfe, CDC Bold and CDC Bow. The reasons for these similarities are due to the large negative loadings of CAT and CAF presence in these cultivars. Newdale, CDC Copeland, Harrington, AAC Goldman and Lowe did not belong to any distinct grouping based on the multivariate data analysis.

A dendrogram was generated by cluster analysis as shown in Figure 4B. The mean values of eight measured phenolic concentrations within ten barley cultivars were used for the analysis. The barley cultivars were clustered based on a similarity of 70%. Different clusters consist of a different number of cultivars which make each cluster markedly different from other clusters observed. The closest cultivars based on the similarity level of 73.71% were AAC Synergy and CDC Mindon due to the presence of FA and PCA. The second closest cultivars were Newdale and AAC Goldman, which had a similarity level of 60.64%. The third closest cultivars were CDC Bow and AC Metcalfe, which had a similarity level of 55.19%. Most neighbouring cultivars within the clusters were closely related to each other. Based on the phenolic compounds present in each cultivar, Newdale, AAC Goldman and Lowe showed a distance from all the other cultivars tested, whereas the Harrington cultivar was markedly different from other studied cultivars. At a similarity level of 50%, CDC Bold, AAC Synergy and CDC Mindon showed higher similarities compared to CDC Bow, AC Metcalfe and CDC Copeland. These data suggest that the differences in the phenolic compound profile led to the obtained similarities in different barley cultivars analyzed in the current study.

### 3.3. Mycotoxin Contents of FHB-Infected Barley Cultivars

We conducted a comparison of disease severity and mycotoxins in FHB-infected cultivars of barley. Fusarium head blight and DON content demonstrated differential reactions across cultivars, in agreement with FHB ratings based on multi-year means (as seen in Appendix A). Susceptible and moderately susceptible cultivars showed greater than twice the DON levels of moderately resistant cultivars. For example, the susceptible check (CDC Bold) contained 5.1 ppm DON content, while the moderately resistant check (CDC Mindon) contained 1.9 ppm DON content. The incidence of FHB was highest in CDC Bold and lowest in the moderately resistant CDC Mindon and Lowe (Figure 5). 

Based on the correlation matrix analysis of total individual phenolic compounds, a positive correlation was observed between DON content (ppm) and the content of total CAT in FHB-infected grains (Table 3). All other phenolic compounds did not show a significant correlation with the DON content. There was a strong positive correlation between FA and IsoFA. In addition, a significant positive correlation was observed between total 4HBA and total VAN A in FHB-infected grains of the tested cultivars.

## 4. Discussion

In the face of a changing climate, the ability of plants to survive and occupy unfavourable environments is key. Fusarium head blight is one of the major reasons for economic losses and reduced quality in barley [1,18,19]. The potential contamination of barley grains by DON is a constant menace to domestic and export markets. For example, for public safety concerns, the brewing and malting industry set limits for DON content in malting barley, and a content exceeding >0.5 mg/kg will most likely result in a rejected sale [17]. Most often to control the disease, several individual or combined means are used such as fungicides, biological agents, cultural practices as well as resistant cultivars. Among those, the use of cultivars with inbuilt resistance to diseases is considered to be the most effective, environmentally friendly and economical method. However, development of FHB-resistant barley cultivars is very challenging. Currently, in barley, the highest level of genetic resistance is moderately resistant. A DON content reduction of 20 to 50% is usually observed in these cultivars [20,21]. 

Phenolic compounds including phenolic acids are secondary metabolites found in plants that are important for human and animal health and cause toxic effects in different micro-organisms, including *Fusarium* species. Thus, the main role credited to these compounds in plant defense mechanisms is their antioxidant properties. In addition, it is believed that they also contribute via the re-enforcement of plant structural components that perform as a mechanical barrier against the pathogen [22]. Out of these, FA, hydroxybenzoic acid, SIN, cinnamic acid and VAN A are the predominant phenolic acids [1,5]. Our current study is a descriptive analysis of the main phenolic compounds in clean and FHB-infected grains of Canadian barley cultivars with various reactions to FHB.

A higher concentration of phenolic acids was found in *Fusarium*-resistant wheat and corn plants compared to susceptible plants [5]. Indeed, in our study, the lowest contents of bound phenolic acid contents were found in the FHB-susceptible barley cultivar CDC Bold, while the highest phenolic acid contents were shown in most of the moderately FHB-resistant cultivars in tested clean barley cultivars. For example, PCA, FA, SIN, CAT, Iso FA and VAN A were the phenolic compounds that were the lowest in clean susceptible cultivars, indicating that these compounds may be used for susceptible cultivar identification. The moderately FHB-resistant clean barley cultivars such as CDC Mindon, Lowe, Harrington and AAC Goldman showed the highest contents of PCA, FA, SIN, Iso FA and CAT, corroborating the aforementioned studies on wheat and corn. The presence of higher contents of these phenolic compounds in moderately resistant clean cultivars compared to susceptible cultivars suggests the potential involvement of the phenolic compounds in barley disease resistance.

The cultivar, species and environmental conditions may determine phenolic contents in cereals through constitutive and induced biosynthesis [8]. In our study, FHB-infected grains showed a higher content of phenolic compounds compared with clean barley similar to other studies in cereals [5]. Despite differential contents and changes in phenolic compounds in tested barley cultivars observed after FHB infection, the principal component analysis showed that CDC Mindon and AAC Synergy cultivars possess similarities compared to the other barley cultivars. In addition, CDC Bold, CDC Bow and AC Metcalfe showed similarities compared to the other cultivars tested. These results suggest distinct groups of susceptible to intermediate and moderately resistant groups in total phenolic contents. However, the susceptible cultivar CDC Bold showed the highest significant increase only in the most abundant PCA and FA contents after FHB infection. This change may be because of the accumulation of these marker compounds as a result of infection, not as an intrinsic content, and therefore it could be a marker of FHB infection rather than a marker of disease resistance. However, this suggestion was not valid for other tested cultivars. For instance, the intermediate cultivar CDC-Copeland showed a significant decrease in both tested markers after infection. The reason for this reduction is not clear, but we speculate that this was due to the biotransformation of these compounds to other compounds that are more relevant to the resistance mechanisms [9,11].

It has been suggested that SIN may have the potential to reduce mycotoxin contamination in food and feed [23]. In our study, total SIN content significantly increased after FHB infection in most of the cultivars tested. In addition, CDC Bold (susceptible cultivar) showed the lowest contents of SIN in both clean and infected grains compared with other cultivars, suggesting that SIN could be another phenolic acid which may have a role in FHB infection and DON accumulation. The moderately resistant cultivars CDC Mindon and AAC Goldman showed the highest endogenous accumulation of bound SIN, suggesting this could be one of the biomarkers that could be involved in FHB resistance. The increase in CAT and Iso-FA in the free phenolic fraction after FHB infection in many cultivars may indicate they are as biomarkers of infection rather than biomarkers of disease resistance. We observed a positive correlation between CAT and the DON content, suggesting the potential connection of CAT in mycotoxin production. Accumulation of phenolics may hold a complex relationship with *Fusarium*-resistance. In response to fungal infection, reactive oxygenated species (ROS) are produced by plants in a coordinated defense response. Flavanol compounds such as CAT are strong antioxidant phytochemicals which may help scavenge ROS and assist in reducing the oxidative stress on the plant. Flavanols may also directly reduce TRI-gene expression in the trichothecene pathway [24]. However, elevated levels of CAT may also be symptomatic of a situation of cultivar–pathogen interaction response, where the host defenses have been hyper-stimulated but the battle has been lost. The pathogen may be triggered by the presence of defensive compounds, and stimulate higher DON production. An increase in CAT in the free phenolic fraction could be a result of cell wall degradation by the pathogen. More *in vitro* and *in vivo* studies in barley may be necessary for practical applications.

The findings of this study could be used in potential cultivar distinguishing factors due to the proven potential associations of the accumulation of phenolic compounds such as SIN, PCA, FA, CAF, CAT, Iso FA and VAN A in different cultivars of barley. In addition, some of these phenolic compounds could potentially be used as biomarkers for the selection and development of barley with inbuilt resistance to FHB and lower mycotoxin accumulation. The development of a time- and cost-efficient way for screening phenolic compounds could further be easily adopted and used in barley breeding programs. For example, use of barley cultivars with inbuilt disease resistance could result in a reduction in the number of pesticide applications by barley farmers. This in turn could reduce the greenhouse gas (GHG) emissions associated with barley production. If only one application per year is removed, it will still have a significant value. Moreover, fungicides have the potential to remain in plants and soil for months, enter natural waters and be toxic to aquatic life [25,26]. Thus, a decline in fungicide use could contribute to a reduction in leaching of chemicals into aquifers, improvement of soil-microbiome health and reduction in GHG emissions generated by fungicide production, transportation and application. For example, in Canada, the emissions associated with pesticide production were reported to account for about 6% of the total emissions for barley production [27].

## 5. Conclusions

This study shows the use of barley endogenous phenolic compounds as potential biomarkers in the selection of cultivars for FHB resistance. The difference in phenolic compounds in clean samples may be related to the resistance level of the cultivar. Phenolic compounds such as SIN in the free phenolics portion might be useful biomarkers in disease resistance due to their variation based on the cultivar resistance to FHB. In addition, CAT and Iso FA may be useful to detect infected grains using the free phenolic fraction. While further work is needed to confirm the results of this study, once that is done, a time- and cost-efficient way for screening phenolic compounds should be developed so that it could be easily adopted and used in barley breeding programs.

## Figures and Tables

**Figure 1 biology-12-01306-f001:**
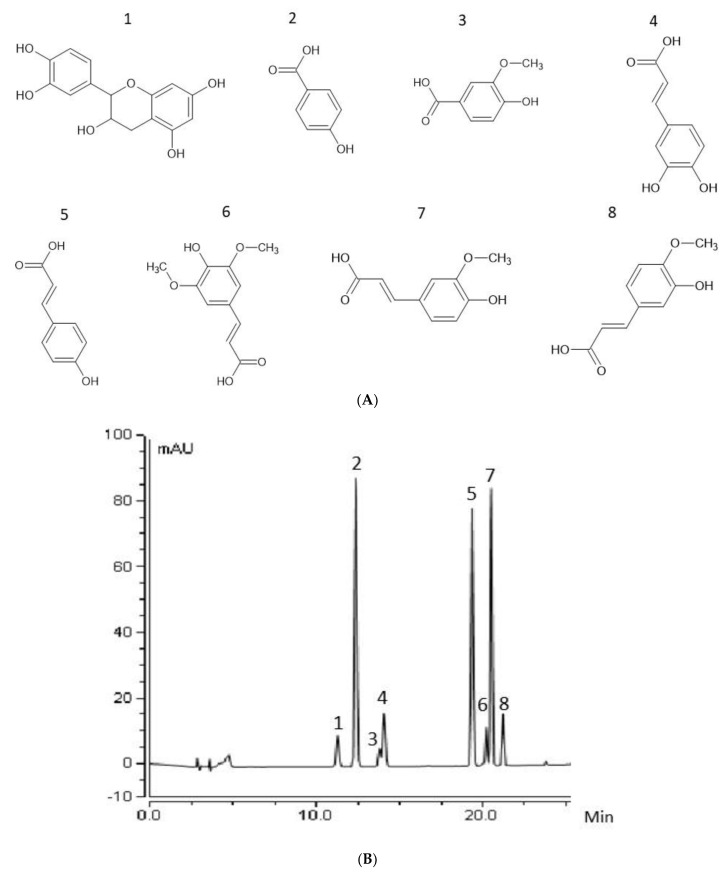
Chemical structures and standards of phenolic compounds quantified in this study. (**A**) Chemical structures of the tested phenolic compounds, (1) catechin (CAT), (2) 4-hydroxybenzoic acid (4HBA), (3) vanillic acid (VAN A), (4) caffeic acid (CAF), (5) para-coumaric acid (PCA), (6) sinapic acid (SIN), (7) ferulic acid (FA) and (8) isoferulic acid (Iso FA). (**B**) Chromatogram of the 8 phenolic standards: (1) RT 11.25, (2) RT 12.35, (3) RT13.79, (4) RT 14.06, (5) RT 19.33 (6) RT 20.2, (7) RT 20.5 and (8) RT 21.2.

**Figure 2 biology-12-01306-f002:**
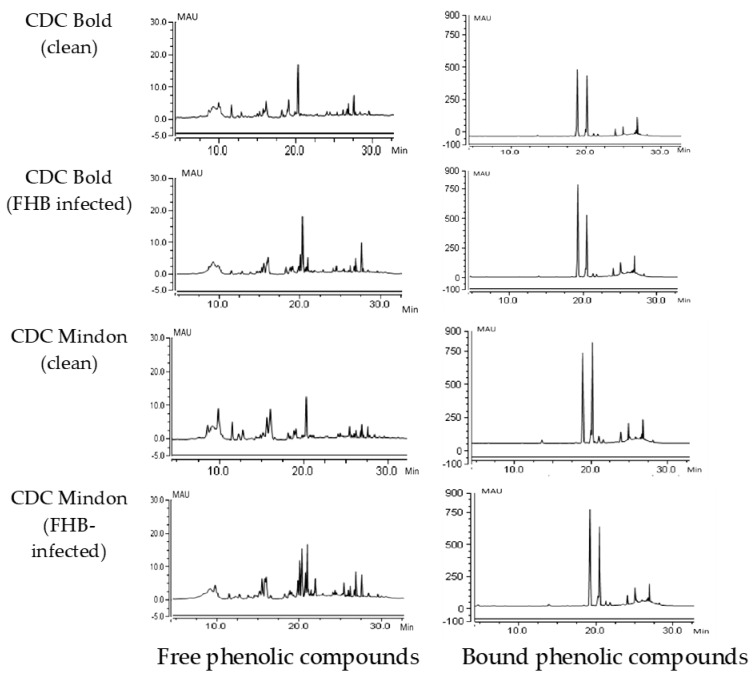
Chromatograms of CDC Bold (susceptible check) and CDC Mindon (resistant check) from the free and bound fractions of clean and FHB-infected barley. Free phenolic compounds, RT 11.0, CAT; RT 18.85, PCA; RT 20.0, SIN; RT 20.15, FA; and RT 21.5, Iso FA. Bound phenolic compounds, RT 11.0, CAT, RT 11.85, 4HBA; RT 13.2, VAN A; RT 13.5, CAF; RT 18.8, PCA; RT 20.0, SIN; RT 20.15, FA, and RT 21.0, Iso FA.

**Figure 3 biology-12-01306-f003:**
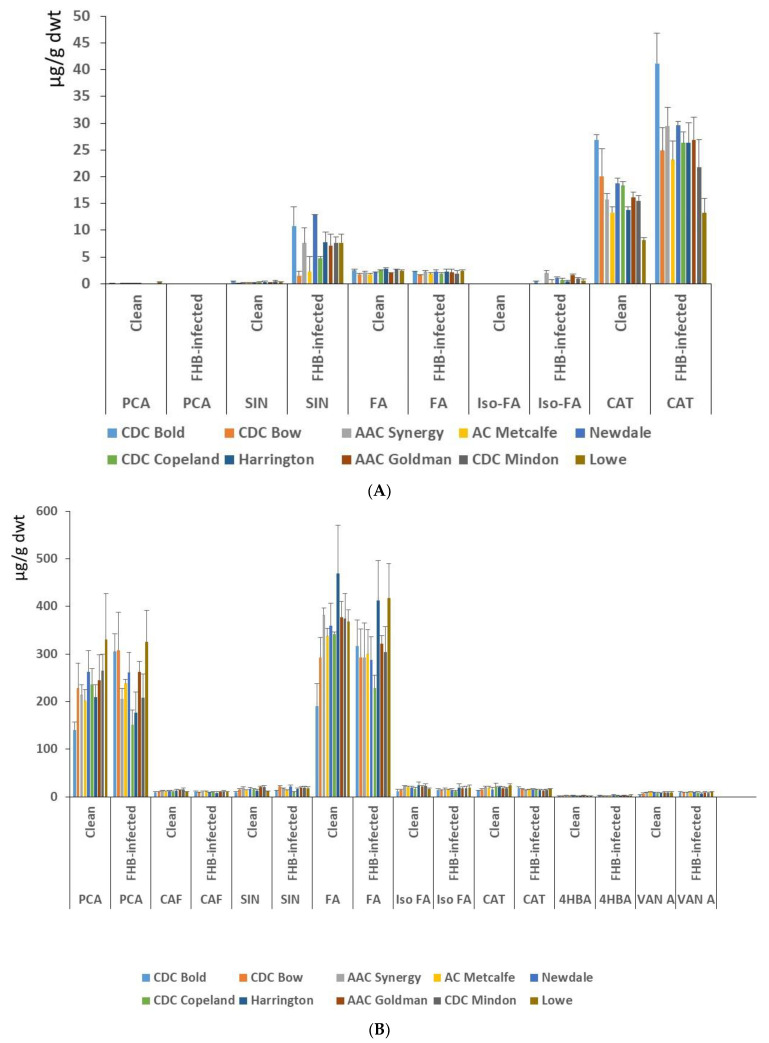
Free (**A**) and bound (**B**) phenolic compounds analyzed by high performance liquid chromatography (HPLC) in non-infected (clean) and FHB-infected grains of the tested barley cultivars. Para-coumaric acid–PCA, sinapic acid–SIN, ferulic acid–FA, isoferulic acid–Iso-FA, catechin–CAT.

**Figure 4 biology-12-01306-f004:**
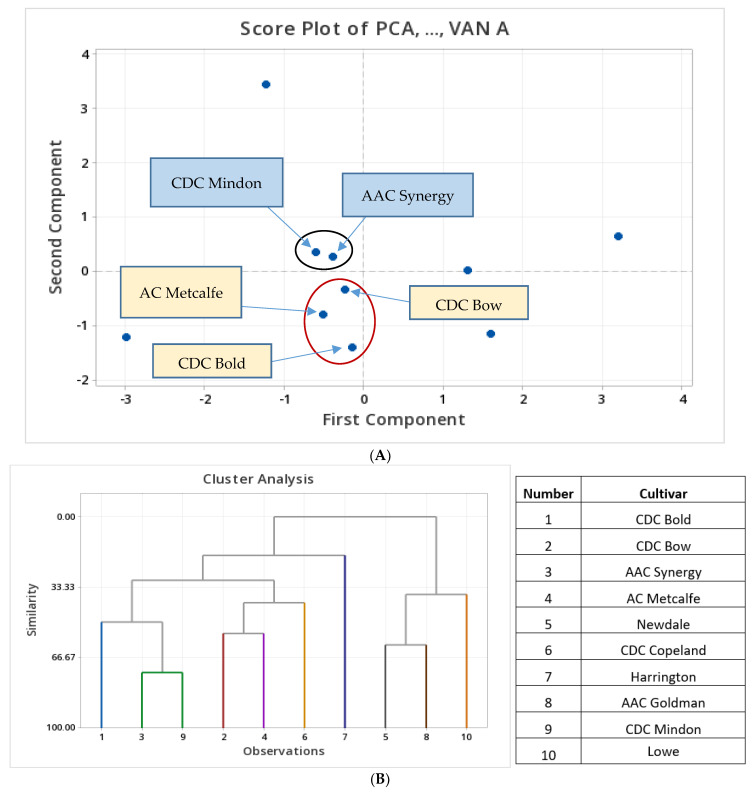
Principal component and cluster analysis of the selected phenolic compounds in ten tested barley cultivars. (**A**) Scree plot of the phenolic compounds from PCA-VAN A. (**B**) Dendrogram generated for the phenolic compounds tested in ten barley cultivars.

**Figure 5 biology-12-01306-f005:**
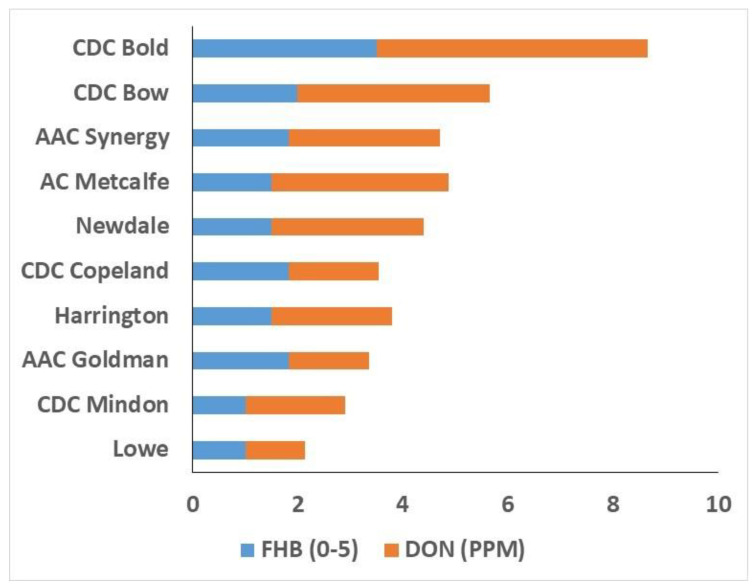
Fusarium head blight (FHB) and deoxynivalenol (DON) content of barley cultivars (*n* = 10).

**Table 1 biology-12-01306-t001:** Pairwise comparison of average bound paracoumaric acid (PCA) content (µg/g dwt) between clean and infected barley.

Cultivar	Clean	Infected	% Change	Significance
CDC Bold	140.5 ± 17	304.5 ± 38	117	0.0024 **
CDC Bow	228.3 ± 53	307.9 ± 80	34.9	ns
AAC Synergy	213.7 ± 22	206.5 ± 22	–3.4	ns
AC Metcalfe	201.5 ± 23	238.1 ± 9.0	18.2	ns
Newdale	262.2 ± 45	260.5 ± 42	–0.6	ns
CDC Copeland	235.9 ± 34	151.9 ± 31	–35.6	0.0337 *
Harrington	208.8 ± 27	176.8 ± 43	–15.3	ns
AAC Goldman	245.1 ± 54	262.6 ± 21	7.1	ns
CDC Mindon	265.3 ± 35	207.7 ± 50	–21.7	ns
Lowe	330.5 ± 96	324.7 ± 67	–1.8	ns

ns: not significant at ≤0.05; * significant at *p* ≤ 0.05; ** significant at *p* ≤ 0.01.

**Table 2 biology-12-01306-t002:** Pairwise comparison of average bound ferulic acid (FA) content (µg/g dwt) between clean and infected barley.

Cultivar	Clean	Infected	% Change	Significance
CDC Bold	191.3 ± 47	317.4 ± 54	65.9	0.03775 *
CDC Bow	292.6 ± 42	292.8 ± 60	0.1	ns
AAC Synergy	382.2 ± 14	292.5 ± 60	–23.5	ns
AC Metcalfe	337.8 ± 15	299.9 ± 52	–11.2	ns
Newdale	359.0 ± 48	288.3 ± 49	–19.7	ns
CDC Copeland	342.4 ± 4.2	227.7 ± 28	–33.5	0.00218 **
Harrington	468.7 ± 101	412.4 ± 85	–12.0	ns
AAC Goldman	377.3 ± 33	321.5 ± 16	–14.8	ns
CDC Mindon	375.3 ± 51.5	304.0 ± 53.5	−19.0	ns
Lowe	367.7 ± 25.5	416.7 ± 73.5	13.3	ns

ns: not significant at ≤0.05; * significant at *p* ≤ 0.05; ** significant at *p* ≤ 0.01.

**Table 3 biology-12-01306-t003:** Correlation matrix showing the relationship between DON content (ppm) and individual total phenolic compounds (µg/g Dwt) in FHB-infected barley cultivars.

	PCA	CAF	SIN	FA	CAT	4HBA	VAN A	IsoFA	DON
PCA		−0.036	0.323	0.314	0.091	0.54	0.712	0.096	0.318
CAF			−0.227	−0.23	−0.062	0.116	0.244	0.136	−0.053
SIN				0.288	0.019	0.534	0.206	0.451	−0.091
FA					−0.313	−0.041	0.006	0.745 *	−0.228
CAT						−0.093	−0.096	−0.459	0.813 **
4HBA							0.792 **	0.05	−0.216
VAN A								0.042	0.048
IsoFA									−0.493
DON									

* Significant at *p* ≤ 0.05, ** significant at *p* ≤ 0.01.

## Data Availability

The data that support the findings of this study are available from the corresponding author upon reasonable request.

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
