# Peer review of "Endogenic Phenolic Compounds of Barley as Potential Biomarkers Related to Grain Mycotoxin Production and Cultivar Selection"

_biology, 2023, doi:10.3390/biology12101306_

Round 1
Reviewer 1 Report
The authors have explored biomarkers for cultivar selection of barley. The biomarkers will be useful to develop new cultivars. Generally, the manuscript is well prepared. However, I have a few comments which should be clarified by the authors prior to publication.
1. The conclusion of the manuscript is not clearly described. The main point of this manuscript is to find the biomarkers to select mycotoxin-resistant cultivars. Therefore, the most important data to be shown would be the difference of phenolic compounds in clean samples related to resistance level of the cultivar. In figure 2A, Sin and CAT might be the useful biomarkers because the level is variant according to the resistance to mycotoxins. On the other hand, PCA, FA would not be good candidates.
2. In table 1, peak annotation is necessary for each chromatogram.
3. In table 2 and 3, uncertainty is necessary.
Author Response
Thank you vey much for the comments. We addressed all the reviewer 1 comments (highlighted in yellow) as below. Please refer the manuscript to see all the changes made.
The authors have explored biomarkers for cultivar selection of barley. The biomarkers will be useful to develop new cultivars. Generally, the manuscript is well prepared. However, I have a few comments which should be clarified by the authors prior to publication.
Thank you very much for your comments.
Q1: The conclusion of the manuscript is not clearly described. The main point of this manuscript is to find the biomarkers to select mycotoxin-resistant cultivars. Therefore, the most important data to be shown would be the difference of phenolic compounds in clean samples related to resistance level of the cultivar. In figure 2A, Sin and CAT might be the useful biomarkers because the level is variant according to the resistance to mycotoxins. On the other hand, PCA, FA would not be good candidates.
A1: Information was added under the conclusion section. Results and discussion parts were also expanded with more explanations. Thank you.
Q2: In table 1, peak annotation is necessary for each chromatogram.
A2: Peak annotations along with the retention times are provided below the table as footnotes.
Q3: In table 2 and 3, uncertainty is necessary
A3: Thank you! Uncertainty (as standard deviation values) has been added to Table 2 and Table 3 as suggested
Reviewer 2 Report
Figures S1 and S 3: units are missing
In material and method section a more detailed information on experimental material is required (Sample dimension, size of fields, methodologies and techniques applied – for examples how was prepared and applied the inoculum, etc.)
Also basic details for rating the Fusarium head blight are encouraged.
Please clearly define the tested materials used and their sampling (the origin of clean and FHB-infected grains is not clear)
Please list the phenolic compounds used as standard, the range of concentration used for quantification and their origin.
Figure 3B, change number with cultivar acronyms
Do authors performed experimental approach to define resistant and non resistant cultivars?
Define the samples identity in terms of cultivar or variety.
L333: corn is correct?
The experimental design should be clearly defined: do authors selected cultivars with different degrees of resistance or they define experimentally the resistance?
Phenolic compounds quantity seems to be increased in presence of FHB, but not in all tested conditions. Do authors propose any hypothesis about this diversity? The increased quantity can be a marker of infection rather than a marker of resistance.
Author Response
Thank you very much for the review. Following are the answers (yellow highlighted) for the comments. Please see the revise manuscript for appropriate changes.
Figures S1 and S 3: units are missing
Units were added for the figures. Thank you!
In material and method section a more detailed information on experimental material is required (Sample dimension, size of fields, methodologies and techniques applied – for examples how was prepared and applied the inoculum, etc.)
More detailed information was added as requested.
Also basic details for rating the Fusarium head blight are encouraged.
The details were added in the text as requested.
Please clearly define the tested materials used and their sampling (the origin of clean and FHB-infected grains is not clear).
Information added in the materials and methods section as well as in Table S1
Please list the phenolic compounds used as standard, the range of concentration used for quantification and their origin.
These information were added under the materials and methods section (section 2.2).
Figure 3B, change number with cultivar acronyms
A table has been attached along with the variety names corresponding to each number.
Do authors performed experimental approach to define resistant and non-resistant cultivars?
Define the samples identity in terms of cultivar or variety.
To address both of above comments, we added the information to the Table S1 (added a column on origin/Breeding program) to define the samples identity, origin resistant and susceptible cultivars, related breeding program etc.
L333: corn is correct?
Yes, corn is correct and included as it is a cereal crop.
The experimental design should be clearly defined: do authors selected cultivars with different degrees of resistance or they define experimentally the resistance?
All the information related to the selected cultivars with different degrees of resistance and their references are available in the modified supplementary table 1 (Table S1) as above.
Phenolic compounds quantity seems to be increased in presence of FHB, but not in all tested conditions. Do authors propose any hypothesis about this diversity? The increased quantity can be a marker of infection rather than a marker of resistance.
The authors agree with this hypothesis. We noticed that there are significant changes in only some cultivars after the FHB infection. For instance, CDC Bold cultivar (Susceptible check) showed a significant increase in the two major compounds ferulic acid and p-coumaric acid contents, which may imply that the accumulation of these marker compounds was induced as a result of infection not as intrinsic content, and therefore it can be a marker of infection rather than a marker of resistance. However, this hypothesis was not valid for other varieties. For instance, the moderately resistant CDC-Copeland showed a significant decrease in both marker compounds after infection. The reason for this reduction is not known, but one can speculate that this was due to the biotransformation of these compounds to other compounds that are more relevant to the resistance mechanisms. More studies (Eg: growing these cultivars in different locations and seasons) may be necessary to confirm. We included this information in discussion section.
Round 2
Reviewer 2 Report
Authors perfomed a deep revision of the manuscript according to all comments exposed by reviewers. The manuscript could be publishged in present form
Author Response
Comments and Suggestions for Authors
Authors performed a deep revision of the manuscript according to all comments exposed by reviewers. The manuscript could be published in present form
Thank you very much for your comment. As there are no specific changes needed for the manuscript, I have attached the clean copy of that.
